# Synergy between Phage Sb-1 and Oxacillin against Methicillin-Resistant *Staphylococcus aureus*

**DOI:** 10.3390/antibiotics10070849

**Published:** 2021-07-13

**Authors:** Kevin Simon, Wolfgang Pier, Alex Krüttgen, Hans-Peter Horz

**Affiliations:** 1Institute of Medical Microbiology, RWTH Aachen University Hospital, 52074 Aachen, Germany; ksimon@ukaachen.de (K.S.); wolfgang.pier@rwth-aachen.de (W.P.); 2Laboratory Diagnostic Center, RWTH Aachen University Hospital, 52074 Aachen, Germany; akruettgen@ukaachen.de

**Keywords:** *Staphylococcus aureus*, MRSA bacteriophage, oxacillin, synergy

## Abstract

Methicillin-resistant *Staphylococcus aureus* (MRSA) is a notorious pathogen responsible for not only a number of difficult-to-treat hospital-acquired infections, but also for infections that are community- or livestock-acquired. The increasing lack of efficient antibiotics has renewed the interest in lytic bacteriophages (briefly phages) as additional antimicrobials against multi-drug resistant bacteria, including MRSA. The aim of this study was to test the hypothesis that a combination of the well-known and strictly lytic *S. aureus* phage Sb-1 and oxacillin, which as sole agent is ineffective against MRSA, exerts a significantly stronger bacterial reduction than either antimicrobial alone. Eighteen different MRSA isolates and, for comparison, five MSSA and four reference strains were included in this study. The bacteria were challenged with a combination of varying dosages of the phage and the antibiotic in liquid medium using five different antibiotic levels and four different viral titers (i.e., multiplicity of infections (MOIs) ranging from 10^−5^ to 10). The dynamics of the cell density changes were determined via time-kill assays over 16 h. Positive interactions between both antimicrobials in the form of facilitation, additive effects, or synergism were observed for most *S. aureus* isolates. These enhanced antibacterial effects were robust with phage MOIs of 10^−1^ and 10 irrespective of the antibiotic concentrations, ranging from 5 to 100 µg/mL. Neutral effects between both antimicrobials were seen only with few isolates. Importantly, antagonism was a rare exception. As a conclusion, phage Sb-1 and oxacillin constitute a robust heterologous antimicrobial pair which extends the efficacy of a phage-only approach for controlling MRSA.

## 1. Introduction

Methicillin-resistant *Staphylococcus aureus* (MRSA) can cause a range of organ-specific infections, the most common being in the skin and subcutaneous tissues followed by invasive infections like osteomyelitis, meningitis, pneumonia, lung abscess, and empyema [1]. In addition, a high morbidity and mortality rate is associated with MRSA bloodstream infections [2]. One major factor for resistance in MRSA is the mecA gene encoding for the penicillin binding protein 2a (PBP2a), which exhibits low binding affinity to beta-lactams conferring resistance against this class of antibiotics, including oxacillin [1]. Given the high number of additional resistance mechanisms against further antibiotic classes, such as tetracyclines, aminoglycosides, and macrolides, the treatment of MRSA infections becomes increasingly challenging. While bacteriophages (briefly phages) have recently been given renewed attention as an antibacterial therapeutic option, the dual approach using phages and antibiotics together holds promising potential for the successful control of multi-drug resistance [3,4]. The precise underlying mechanisms of positive or negative phage–antibiotic interactions at the molecular level remain to be discovered in many cases. However, one can generally expect that two differently acting selective pressures are more difficult for the bacteria to overcome than a single antibacterial agent [3]. For instance, *S. aureus* forms aggregates when challenged with gentamicin, whereby the cells become more susceptible to phage predation by the increased number of cell receptors [5]. Furthermore, the development of phage resistance by modifying multi-drug efflux pump activity can increase the sensitivity to antibiotics [6]. A number of studies have already addressed the potential impact of a dual approach against *S. aureus* using a lytic phage in combination with MRSA-effective antibiotics, such as rifampicin, linezolid, fosfomycin, daptomycin, and vancomycin [7,8,9,10,11,12,13], as well as in combination with aminoglycosides, quinolone antibiotics, and β-lactam antibiotics [5,11,12,13]. Interestingly, synergistic interactions between a phage and an antibiotic can also be observed in cases in which the pathogen is resistant against the antibiotic [14,15]. However, combining a phage with an antibiotic, which as a sole agent does not work against MRSA, has not been investigated thus far. In addition, when investigating phage–antibiotic synergy, many studies rely on single or few reference strains or clinical isolates only as being representative for the whole species. However, given the genetic and phenotypic heterogeneity of opportunistic pathogens and even laboratory strains [16,17,18,19], it is important to test the suitability of potential phage–antibiotic pairs on a larger set of bacterial isolates. In this study, we tested the hypothesis that oxacillin, the antibiotic of choice against Methicillin-susceptible *S. aureus* (MSSA), and phage Sb-1, a member of the Twort-like viruses, exert stronger activity against MRSA in combination than either antimicrobial substance alone. Phage Sb-1 has a broad host range, and due to its potential suitability for phage therapy, its biology and genomic features have been well studied over the years [20,21,22,23]. However, *S. aureus* can elude the entire phage-based clearance by several mechanisms (e.g., the formation of small-colony variants) [13]. Phage therapy would therefore greatly benefit from the possibility of co-treatment of a MRSA infection with an antibiotic. We included a panel of 18 distinct clinical MRSA and five MSSA isolates along with four reference strains of *S. aureus* in this study. The bacteria were challenged with varying dosages of oxacillin and phages, both alone and in combination using an optically based microtiter plate assay system.

## 2. Results and Discussion

Twenty-three clinical isolates (18 MRSA and 5 MSSA), designated with the abbreviation “SA” and consecutive numbering, and the four *S. aureus* reference strains ATCC 14154, ATCC 25923, ATCC 27660, and ATCC 29213 were included in this study. While otherwise exhibiting consistent resistance profiles, variable sensitivities for Levofloxacin, Gentamicin, Clindamycin, Co-trimoxazole, Tetracycline, Erythromycin, and Mupirocin allowed the categorization of the 18 MRSA isolates into six resistance patterns (pattern 1: SA1, SA3, SA8, SA11, SA14, SA16, SA18, and SA20; pattern 2: SA5; pattern 3: SA6, SA9, SA12, SA15, and SA19; pattern 4: SA4 and SA13; pattern 5: SA7; pattern 6: SA17 (Appendix A)). A further differentiation of the isolates was possible via ERIC-PCR, which ruled out the existence of duplicates (Appendix A).

Phage Sb-1 was isolated from the commercially available phage cocktail “Staphylococcal Bacteriophage”, which was obtained from the Eliava Institute of Bacteriophages, Microbiology, and Virology (Tbilisi, Georgia) using a clinical *S. aureus* isolate as the propagation host. Phage Sb-1 was able to lyse 16 of the 18 MRSA isolates and 4 of the 5 MSSA isolates. The MSSA strain SA2 and the MRSA strains SA7 and SA17 were resistant to the phage. Furthermore, all four reference strains ATCC 14154, ATCC 25923, ATCC 27660, and ATCC 29,213 were susceptible to phage Sb-1.

Bacteria as planktonic cultures were challenged with phages at four different multiplicities of infection (i.e., MOI 10, 10^−1^, 10^−3^, and 10^−5^), either alone or in combination with oxacillin. Five different oxacillin concentrations in the range below and above the MICs of the propagation strain SA19 (i.e., 100 µg/mL, 50 µg/mL, 20 µg/mL, 10 µg/mL, and 5 µg/mL) were used. We classified the extent of bacterial suppression into five categories according to Chaudhry et al., 2017 [24]: (1) synergism, where the combined treatment leads to a stronger bacterial reduction than the sum of either substance alone; (2) additive, where the combined treatment leads to a bacterial reduction which equals the sum of either substance alone; (3) facilitation, where the combined treatment leads to a stronger bacterial reduction than the best-acting single agent but less than the additive level; (4) neutral, where the combined treatment leads to a bacterial reduction that equals the best-acting single agent; and (5) antagonism, where the combined treatment is worse than the best-acting single agent.

For illustration, the dynamics of the cell density changes in the time-kill assays are representatively shown for the MRSA strain SA19. As expected, oxacillin alone only moderately suppressed the bacteria. A stronger antibacterial activity was achieved with the phage; however, the co-addition of oxacillin led to the strongest bacterial decline (Figure 1a).

Depending on the MOI, the antimicrobial effect of the combination approach was either synergistic or additive, causing stronger bacterial reduction with phage MOIs of 10 and 0.1 compared with the lower phage titers (Figure 1b). Appendix A depicts a heat map for all 27 *S. aureus* isolates, each of which challenged with 30 different phage/oxacillin ratios. For most MRSA strains, oxacillin alone, even at highest level, did not lead to any considerate bacterial suppression (white or light green fields at the right side of the diagrams in Appendix A). With few exceptions, significant positive interactions became particularly evident with phage MOIs of 10 and 0.1 (green fields at the left side of the diagrams in Appendix A). This was true for MRSA and MSSA, whereby in the latter case, phage activity was also apparent more often with lower MOIs. Examples for synergism, additive effects, and facilitation are representatively highlighted in Appendix A for the strains SA9, SA13, and SA ATCC 25923, respectively, for 5 µg/mL oxacillin and a phage MOI of 0.1. Likewise, an example of a neutral effect and antagonism is highlighted for strains SA12 and SA21, respectively, in this figure. In most cases, the co-addition of 5 µg/mL oxacillin already accounted for the bulk of the bacterial suppression. Although higher oxacillin levels also led to positive interactions with the phage, they did not necessarily imply an excess therapeutic value (Appendix A). The efficiency of the phage alone in reducing the bacteria varied largely from strain to strain. Particularly when the phage alone did not perform so well, the benefit of the combined approach was most apparent, as it complemented the lack of phage-based bacterial reduction (Appendix A). For instance, the phage alone suppressed strain SA22 only by about 35%. However, with the combined approach, SA22 was suppressed by about 90%, (which was a gain of around 55%; Appendix A). Conversely, SA19 was already suppressed by the phage very efficiently, so that co-addition of oxacillin contributed only a gain of a few percentage points to the bacterial suppression (Appendix A).

Overall, the positive interactions were dominated by synergism (observed with 11–15 strains, depending on the antibiotic concentration), followed by additive effects and facilitation seen in up to six and seven strains, respectively (Figure 2 and Table 1).

Neutral effects occurred in up to five strains. Importantly, antagonism was observed in only one case (SA21) (Figure 2 and Table 1). Even for this strain, the extent of antagonistic interactions was comparably weak (compare in Appendix A). Hence, for virtually all strains tested in this study, co-administration of oxacillin could be recommended as an adjunct to Sb-1 during phage therapy, as it likely would improve rather than worsen the clinical outcome. Table 1 also shows that the mode of interaction was consistent for many strains over the tested oxacillin range. For instance, synergism could be seen throughout for strains SA7, SA9, SA11, SA16, SA18, SA20, SA24, and SA ATCC 27660, irrespective of the oxacillin levels. For the other strains, a switch to a “better” interaction category could be observed with descending antibiotic levels (e.g., strain SA1 switching from additive to synergism or strain SA3 switching from facilitation to additive).

In summary, the data indicate that a bactericidal antibiotic does not necessarily jeopardize the advantages of combination therapies. However, we are currently unaware of the precise molecular mechanisms that lead to the collaborative effect. In our ongoing studies, we are therefore elucidating the nature of those phage–antibiotic interactions based on transcriptome analysis. Apparently, the antibiotic does not impede the phage’s capacity to prey on his bacterial host. In fact, Sb-1-like phages isolated from the traditional Georgian Pyo-Cocktail [25] have been shown to replicate in the presence of oxacillin [12,13]. However, Berryhill et al. observed an enhanced antibacterial effect only when the phage and oxacillin were given sequentially to the MSSA strain “Newman”, which was not tested in our study, as opposed to simultaneous administration [13]. It is therefore likely that the suppression of the clinical isolates, which were tested in in our study, could be further enhanced when the two antimicrobials had been administered in a time-delayed manner. To date, phage therapy is still not approved in most countries, but more and more clinical trials have been launched in recent years. In the meantime, we are learning from a growing number of phage applications in the form of compassionate treatment [26]. To this end, phage therapy is usually accompanied with an antibiotic whereby, naturally, oxacillin is not considered a suitable adjuvant for a phage-based treatment of an MRSA infection [26]. However, our findings—based on a number of different MRSA strains—suggest that oxacillin could in fact improve the clinical outcome of phage-based treatment of MRSA infections. Therefore, continuative studies are warranted to verify the therapeutic value of both antimicrobials in vivo. One should also bear in mind that beta-lactam agents are relatively safe and well-tolerated compared with other antibiotic classes [27] and that the use of last resort antibiotics should be kept to a minimum. The most common anti-MRSA antibiotics are vancomycin, daptomycin, ceftaroline, and linezolid. However, their frequent use has accelerated drug resistance development even against those antibiotics [28]. As a conclusion, phage Sb-1 alone already has a proven record for the treatment of *S. aureus* infections in humans [23]. However, Sb-1-based suppression of MRSA can be significantly fostered by the putatively useless antibiotic oxacillin. This means that the efficacy of old antibiotics can be preserved with phages, which enlarges treatment options and gives hope that we are regaining some lost time in the fight against pan-resistant superbugs.

## 3. Methods

### 3.1. Description of S. aureus Strains

A total of 23 clinical isolates of *S. aureus* (18 MRSA and 5 MSSA), obtained from various clinical specimen at the University Hospital RWTH Aachen, were used. Species identity of these isolates (designated as SA01–SA24) could be confirmed via MALDI-TOF mass spectrometry (Microflex LT, Bruker Daltonik GmbH, Bremen, Germany), except for SA10, which was disregarded for further studies. The automated antibiotic susceptibility testing of clinical isolates, including the detection of the *mecA* gene, was performed using the VITEK2 system (bioMérieux, Marcy-l’Étoile, France). In addition, four *S. aureus* reference strains (ATCC 14154, ATCC 25923, ATCC 27660, and ATCC 29213) were included in this study. Genomic distinctiveness of all isolates was verified based on enterobacterial repetitive intergenic consensus PCR (ERIC-PCR) as described previously [29].

### 3.2. Description of Phage Sb-1

Phage Sb-1 was isolated from the commercially available phage cocktail “Staphylococcal Bacteriophage”, which was obtained from the Eliava Institute of Bacteriophages, Microbiology, and Virology (Tbilisi, Georgia) using a clinical *S. aureus* isolate as the propagation host. The phage lysate was purified by three consecutive single-plaque isolations and propagations. For short-term storage (e.g., several weeks), the phage lysates were stored at 4 °C. For long-term storage, the phages were mixed with glycerol (20% (*V/V*)) in equal parts and stored in CryoPure tubes (Sarstedt, Nuembrecht, Germany) at −196 °C in liquid nitrogen.

Phage identity was confirmed by whole genome sequencing using the MiSeq platform as described previously [14]. The isolated phage shared a genome-wide nucleotide identity of 99.90% with the reference genome of phage Sb-1 (HQ163896) [20] and an identity of 99.99% with phage Sb-1_8383 (accession-no: MN336261.1) [23].

### 3.3. Time-Kill Assays

Infection of planktonic bacterial cells with a starting amount of approximately 5 × 10^8^ cfu/mL was done at the exponential phase of bacterial growth in Lysogeny broth (LB) (NaCl 1% *w*/*v*, tryptone 1% *w*/*v*, yeast extract 0.5% *w*/*v*). The bacteria were challenged with phages at four different multiplicities of infection (MOI 10, 10^−1^, 10^−3^, and 10^−5^), either alone or in combination with oxacillin. Five different oxacillin concentrations in the range below and above the MICs of the propagation strain SA19 (100 µg/mL, 50 µg/mL, 20 µg/mL, 10 µg/mL, and 5 µg/mL) were used, which largely reflected achievable serum levels in human anti-infective therapy [30,31]. The dynamics of phage- or antibiotic-based challenges on bacterial populations were determined via measurements of the changes in optical densities (OD_590_), which is particularly suited for high throughput analysis [32,33,34]. According to our previous studies, the presence of a β-lactam antibiotic does not interfere with the optical density values, which correlated with the reduction of bacterial cells rather than with altered cell morphology [14,15].

For the infection assays, 10 mL of 2× LB were inoculated with 100 µL of the host strain and incubated overnight at 37 °C and 200 rpm. Subsequently, 5 mL of this culture was again mixed with 5 mL 2× LB and shaken at room temperature for another hour to reach a bacterial concentration of approximately 5 × 10^8^ CFU/mL. The phage-lysates were diluted by tenfold serial dilution, and 100 µL of phage-lysates were inoculated with 98 µL of a bacterial strain and 2 µL of the selected antibiotic on a 96-well microtiter plate. For the controls, 98 µL of the host strain or LB medium were mixed with 2 µL of RNase-free water and 100 µL PBS. After sealing the microtiter plate with adhesive tape, a hole was made in every well to supply the bacteria with oxygen throughout the experiment. The time-kill assays were run for 16 h at 37 °C, and the OD_590_ was measured every 20 min using the SpectraMax i3 microplate reader (Molecular Devices, Sunnyvale, CA, USA).

### 3.4. Data Analysis

The OD_590_ data were used for calculating the area under the curve (AUC) via numerical integration with the formula $\overset{48}{\underset{i=0}{\sum }}\frac{f\left(i\times \Delta t\right)+f\left(i+1\times \Delta t\right)}{2}\times \Delta t$, with Δt=20 min=0.33 h and f(i×Δt) representing the OD_590_ values measured every 20 min for 16 h [14]. The AUC of the treated bacteria was related to the AUC of the bacterial controls in order to calculate the extent of bacterial reduction. The extent of bacterial suppression was categorized as synergism, additive effect, facilitation, neutral effect, or antagonism, as described previously [24]. All assays were run in triplicate. Statistical analysis (phage–oxacillin combinations versus the phage alone approach) was performed using a two-tailed *t*-test with a significance level of *p* < 0.05.

## Figures and Tables

**Figure 1 antibiotics-10-00849-f001:**
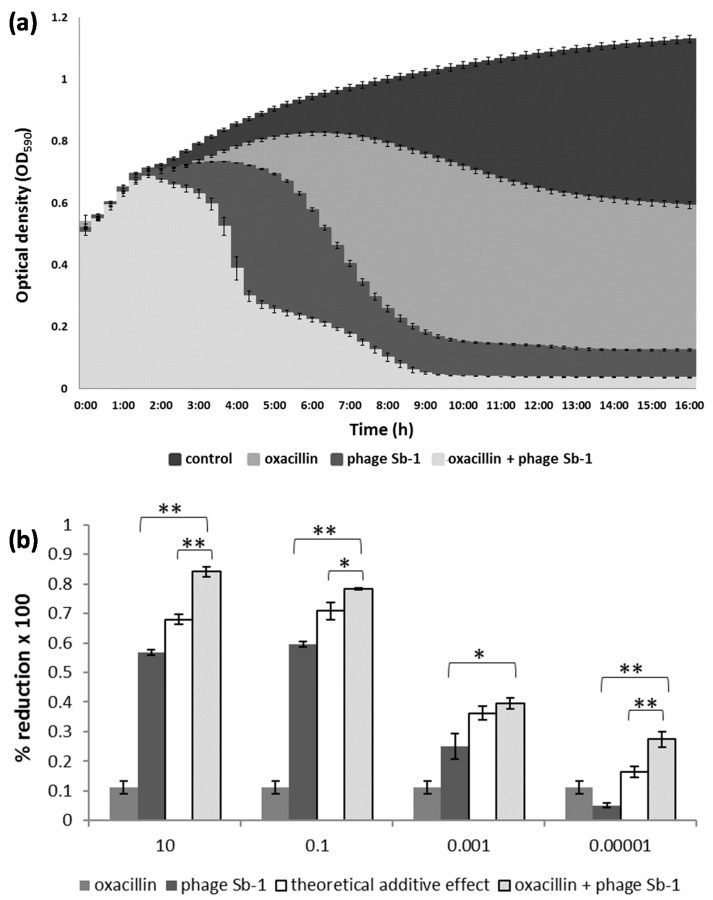
Interaction between phage Sb-1 and oxacillin against the MRSA clinical isolate SA19. (**a**) The mean bacterial density (OD_590_) of three biological replicates over time is shown using the phage at an MOI of 0.1 and oxacillin at a concentration of 10 µg/mL in liquid medium. (**b**) The percentage of bacterial reduction based on the area under the curve (AUC) of the treated bacteria in relation to the AUC of the untreated bacteria, using phage MOIs ranging from 10 to 0.00001. The bars represent the mean reduction of each treatment for three biological replicates ± standard deviations. The stars indicate statistical support for an enhanced antibacterial effect of the dual approach in comparison with the effect of the phage alone or with the theoretical additive effect. ** *p* < 0.01. * *p* < 0.05.

**Figure 2 antibiotics-10-00849-f002:**
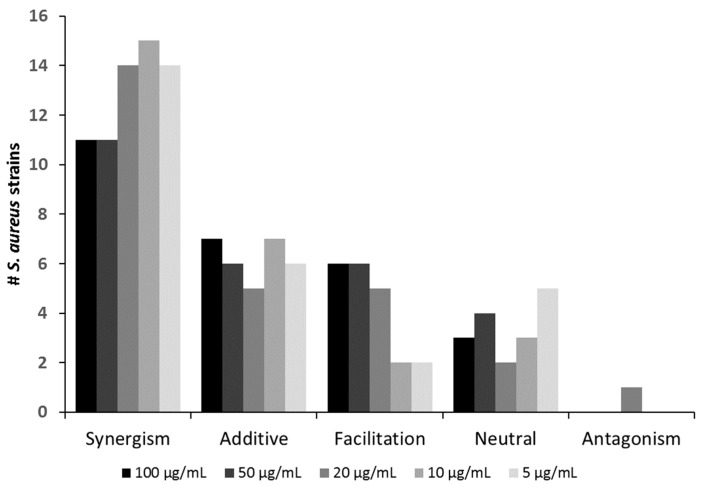
Number of *S. aureus* strains against which the phage Sb-1 (MOI 0.1) interacted in a positive, neutral, or antagonistic fashion with the antibiotic oxacillin at varying dosages.

**Table 1 antibiotics-10-00849-t001:** List of *S. aureus* strains against which the phage Sb-1 (MOI = 0.1) interacted positively or negatively with oxacillin, depending on the antibiotic concentration.

Oxacillin (µg/mL)Effect	100	50	20	10	5
Synergism	SA7, SA9, SA11, SA12, SA15, SA16, SA18, SA20, SA22, SA24,SA ATCC 27660	SA7, SA9, SA11, SA12, SA15, SA16, SA18, SA19, SA20, SA24,SA ATCC 27660	SA1, SA6, SA7, SA8, SA9, SA11, SA12, SA15, SA16, SA18, SA19, SA20, SA24,SA ATCC 27660	SA1, SA6, SA7, SA8, SA9, SA11, SA15, SA16, SA17, SA18, SA19, SA20, SA22,SA24,SA ATCC 27660	SA1, SA6, SA7, SA8, SA9, SA11, SA16, SA18, SA19, SA20,SA24,SA ATCC14154,SA ATCC 27660,SA ATCC 29132
Additive	SA1, SA6, SA8,SA13, SA14,SA19, SA23	SA1, SA6, SA8,SA13, SA14,SA23	SA3, SA4, SA5,SA13, SA23	SA3, SA4, SA5,SA13, SA23,SA ATCC 14154,SA ATCC 29132	SA3, SA4, SA5,SA13, SA15,SA23
Facilitation	SA3, SA4, SA5,SA ATCC 14154,SA ATCC 25923,SA ATCC 29132	SA3, SA4, SA5,SA ATCC 14154,SA ATCC 25923,SA ATCC 29132	SA14, SA17,SA ATCC 14154,SA ATCC 25923,SA ATCC 29132	SA14,SA ATCC 25923	SA14,SA ATCC 25923
Neutral	SA2, SA17, SA21	SA2, SA17, SA21, SA22	SA2, SA22	SA2, SA12, SA21	SA2, SA12, SA17, SA21, SA22
Antagonism			SA21

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
