# Peer review of "Synergy between Phage Sb-1 and Oxacillin against Methicillin-Resistant Staphylococcus aureus"

_antibiotics, 2021, doi:10.3390/antibiotics10070849_

Round 1

Reviewer 1 Report

This manuscript describes a study of examining the synergistic effect of phage Sb-1 and oxacillin against MRSA. This is an interesting and straight forward study. However, the manuscript should be polished, and the writing can be clearer. One major recommendation for the authors is that better definitions of “synergism”, “addictive”, “facilitation”, “neutral” and “antagonism” should be provided. It is not difficult to understand “synergism” and “antagonism” but how are the other three different from each other? In the 2nd paragraph on page 4, it says “neutral effects which means that the dual approach was not better that the best acting single agent”. Should “antagonism” be the same case too?

Other comments:

  1. Abstract: first sentence should be “for not only a number….but also infections”.
  2. Abstract: sentence “Bacteria were challenged…” should be written.
  3. Abstract: “bastion”?
  4. Abstract: “phage titers of 10^-1 and 10”…did you mean “titers with MOI of 10^-1 and 10”?
  5. Introduction: “high morbidity and mortality” means “high morbidity and mortality rate”?
  6. Introduction: what does “further antibiotic classes” mean?
  7. Introduction: “an” is missing in the middle of “as antibacterial therapeutic”.
  8. Introduction: check the singular and plural forms for words in the sentence “Interestingly, synergistic …”.
  9. Page 3: rewrite “consistently evident particularly”?
  10. Page 4: top, rewrite “In most cases…value”.
  11. Page 4: Antagonism is the scenario that “a bactericidal antibiotic does not necessarily jeopardize the advantages of combined therapies”. Should it be the opposite or my understanding regarding antagonism is incorrect?

Author Response

Reviewer 1:

This manuscript describes a study of examining the synergistic effect of phage Sb-1 and oxacillin against MRSA. This is an interesting and straight forward study. However, the manuscript should be polished, and the writing can be clearer. One major recommendation for the authors is that better definitions of “synergism”, “addictive”, “facilitation”, “neutral” and “antagonism” should be provided. It is not difficult to understand “synergism” and “antagonism” but how are the other three different from each other? In the 2nd paragraph on page 4, it says “neutral effects which means that the dual approach was not better that the best acting single agent”. Should “antagonism” be the same case too?

Author: We thank Reviewer 1 very much for his helpful comments. We have written certain parts in the manuscript clearer, in particular with emphasis on the definitions of the concepts synergism, additive, facilitation, neutral and antagonism. Other comments:

  1. Abstract: first sentence should be “for not only a number….but also infections”.

Author: This has been changed.

  1. Abstract: sentence “Bacteria were challenged…” should be written.

Author: The sentence already reads “Bacteria were challenged”

  1. Abstract: “bastion”?

Author: This word has been replaced by „antimicrobials“

  1. Abstract: “phage titers of 10^-1 and 10”…did you mean “titers with MOI of 10^-1 and 10”?

Author: Yes, the word “titer” has been replaced by “MOI”.

  1. Introduction: “high morbidity and mortality” means “high morbidity and mortality rate”?

Author: Yes, this has been changed in the manuscript.

  1. Introduction: what does “further antibiotic classes” mean

Author: For clarity, we have added the names of a few additional antibiotic classes into the manuscript.

  1. Introduction: “an” is missing in the middle of “as antibacterial therapeutic”.

Author: This has been corrected.

  1. Introduction: check the singular and plural forms for words in the sentence “Interestingly, synergistic …”.

Author: We have corrected this sentence.

  1. Page 3: rewrite “consistently evident particularly”?

Author: This sentence has been rephrased.

  1. Page 4: top, rewrite “In most cases…value”.

Author: This sentence has been modified and split into two sentences for clarity.

  1. Page 4: Antagonism is the scenario that “a bactericidal antibiotic does not necessarily jeopardize the advantages of combined therapies”. Should it be the opposite or my understanding regarding antagonism is incorrect?

Author: Thank you for this comment. This sentence in the manuscript was misleading, and has now been modified for clarity.

Reviewer 2 Report

The manuscript antibiotics-1266159 was submitted as brief reports. The report is brief and in my opinion should be developed in the experimental part.
The introduction should also be broadened. Authors cite only 24 references which is a small amount in this field.

Author Response

Reviewer 2:

The manuscript antibiotics-1266159 was submitted as brief reports. The report is brief and in my opinion should be developed in the experimental part.
The introduction should also be broadened. Authors cite only 24 references which is a small amount in this field.

Author: We thank Reviewer 2 for his helpful comments. We have incorporated more text into the manuscript, in particular by explaining in more detail the experimental part at the beginning of the result and discussion section and by moving supplemental method information into the methods part of the main manuscript. We have also extended the introduction part along with increasing the overall number of references.

Reviewer 3 Report

The manuscript by Simon et al describes the possibility of using a combined treatment of phage Sb1 and and oxacillin to fight S.aureus MRSA infections. They used 17 MRSA strains and as controls 5 MSSA and 4 references ones.

The interest of finding alternatives in the treatment to the resistant strains is beyond doubt. And all the efforts made in the way are welcomed and of great interest for the research community.

This paper describes the sensibility of the used strains to the phage infection and 8 antibiotics (VITEK2 system) and an additional one unique experiment of bacteria reduction assays using different amounts of oxacillin and several phages titers and given the data as AUC (area under the curve). All the figures presented figures (except the Fig S1) are based in this experiment.

The experiment is well designed and performed. The methods are  explained enough. But I have found several flaws in the manuscript writing:

  • The introduction does not explain the mechanism by the interactions of these two antimicrobials should be an improvement (the target of oxacillin is not present or its weak in its interaction) or a theory to that.
  • The results  and discussion must be improved in general.
  • An introduction to the experiment should be added.
  • More details of the strains given examples of the different behavior. 1) Mention the strains not susceptible to phage   2)18 MRSA were categorized in 6 groups, ¿which ones? say the strains corresponding to each group in the text and mark the groups in the figure S1; 
  • Explain the criteria for the different phenotypes: additive, synergistic and facilitation.
  • Explain in the text with more detail the Fig S2. Which are the strains and conditions for each phenotype, at least an example of each one to understand better the text.
  • A table with the strains corresponding to the data presented in the figure 2 should be also included to verify if the authors perception is the same as the reviewers related to phenotypes.
  • Neutral effects: which strains?
  • Antagonism: which one?
  • The Fig S3 is not well explained neither in the text nor in the legend. The correspondence of the filled and empty diamonds to the strains is nor clear and the name of the strains should be added in them to be able to correlate these data to the one presented in Fig. S2.
  • There is not a clear conclusion in the manuscript

MInor issues

  • The verb tense used should be past (EX. Phage Sb-1 was able..., references strains were susceptible...)
  • Figure 1a: oxacillin not ocacillin
  • Page 4: "... tested in in our study,..." In is duplicated
  • The species in the bibliography should be in italics

Author Response

Reviewer 3:

The manuscript by Simon et al describes the possibility of using a combined treatment of phage Sb1 and and oxacillin to fight S.aureus MRSA infections. They used 17 MRSA strains and as controls 5 MSSA and 4 references ones.

The interest of finding alternatives in the treatment to the resistant strains is beyond doubt. And all the efforts made in the way are welcomed and of great interest for the research community.

This paper describes the sensibility of the used strains to the phage infection and 8 antibiotics (VITEK2 system) and an additional one unique experiment of bacteria reduction assays using different amounts of oxacillin and several phages titers and given the data as AUC (area under the curve). All the figures presented figures (except the Fig S1) are based in this experiment.

The experiment is well designed and performed. The methods are  explained enough. But I have found several flaws in the manuscript writing:

Author: We thank reviewer 3 very much for his helpful comments. Our point-by-point response is given below and we hope that we have addressed all comments adequately.

  • The introduction does not explain the mechanism by the interactions of these two antimicrobials should be an improvement (the target of oxacillin is not present or its weak in its interaction) or a theory to that.

Author: We have included a paragraph into the introduction where we address possible mechanisms of phage/antibiotic interactions.

  • The results  and discussion must be improved in general.
  • Author: The result and discussion section has been improved by providing more information about the experiments and by describing in more details the results.
  • An introduction to the experiment should be added.
  • Author: We have now incorporated explanatory text at the beginning of the results and discussion section for more clarity regarding the experimental design.
  •  
  • More details of the strains given examples of the different behavior. 1) Mention the strains not susceptible to phage   2)18 MRSA were categorized in 6 groups, ¿which ones? say the strains corresponding to each group in the text and mark the groups in the figure S1; 

Author: We have added this information into the text, and we have marked the six different resistance profiles of the 18 MRSA isolates. Note, that we have replaced the word “group” by “resistance pattern”, as some pattern refer only to a single isolate, making the term “group” less appropriate.

  • Explain the criteria for the different phenotypes: additive, synergistic and facilitation.
  • Author: Those terms have now been explained in more detail in the results and discussion section.
  •  
  • Explain in the text with more detail the Fig S2. Which are the strains and conditions for each phenotype, at least an example of each one to understand better the text.
  • Author: Figure S2 has now been explained in more details. For each of the five categories synergism, additive effect, facilitation, neutral effect, and antagonism, we have now highlighted a particular phage/antibiotic concentration in five representative strains in Fig. S2. During revision, we have also identified some minor calculation errors in Fig. S2, which we have now corrected.
  • A table with the strains corresponding to the data presented in the figure 2 should be also included to verify if the authors perception is the same as the reviewers related to phenotypes.
  • Neutral effects: which strains?
  • Antagonism: which one?
  • Author: A table listing all strains corresponding to Figure 2 and the different phenotypes has now been incorporated into the manuscript. By doing so, we noticed a minor error in Figure 2, regarding the number of strains for the different categories. This has now been corrected.
  • The Fig S3 is not well explained neither in the text nor in the legend. The correspondence of the filled and empty diamonds to the strains is nor clear and the name of the strains should be added in them to be able to correlate these data to the one presented in Fig.

Author: We have included the names of all strains into this Figure, for the first diagram, representatively, which refers to the oxacillin concentration of 100 µl/ml. For better visualization this diagram has been enlarged. The correspondence of the strains to the diamonds in the other diagrams is identical, however in order to maintain the overall clarity, the strain names have not been included in the other diagrams. In addition Fig. S3 has also been explained in more detail in the main text.

  • There is not a clear conclusion in the manuscript
  • Author: We believe, that the manuscript does have a conclusion. However, for more clarity, the final sentences of the manuscript have been slightly modified, accordingly.

MInor issues

  • The verb tense used should be past (EX. Phage Sb-1 was.., references strains weresusceptible...)
  • Figure 1a: oxacillin not ocacillin
  • Page 4: "... tested in inour study,..." In is duplicated
  • The species in the bibliography should be in italics

Author: All minor issues have been addressed as suggested.

Round 2

Reviewer 2 Report

I accept in corrected form.

Author Response

We thank the reviewer very much for accepting the corrected Version.